# Enhanced Electrochemical Performance of Li_1.27_Cr_0.2_Mn_0.53_O_2_ Layered Cathode Materials via a Nanomilling-Assisted Solid-state Process

**DOI:** 10.3390/ma12030468

**Published:** 2019-02-03

**Authors:** Chengkang Chang, Jian Dong, Li Guan, Dongyun Zhang

**Affiliations:** 1School of Materials Science and Engineering, Shanghai Institute of Technology,100 Haiquan Road, Shanghai 201418, China; theanswer0328@gmail.com (J.D.); lguan@sit.edu.cn (L.G.); 2Shanghai Innovation Institute for Materials, Shanghai University, Shanghai 200444, China

**Keywords:** cathode material, X-ray diffraction, Cr^3+^/Cr^6+^ redox pairs, specific capacity, cycling performance

## Abstract

Li_1.27_Cr_0.2_Mn_0.53_O_2_ layered cathodic materials were prepared by a nanomilling-assisted solid-state process. Whole-pattern refinement of X-ray diffraction (XRD) data revealed that the samples are solid solutions with layered α-NaFeO_2_ structure. SEM observation of the prepared powder displayed a mesoporous nature composed of tiny primary particles in nanoscale. X-ray photoelectron spectroscopy (XPS) studies on the cycled electrodes confirmed that triple-electron-process of the Cr^3+^/Cr^6+^ redox pair, not the two-electron-process of Mn redox pair, dominants the electrochemical process within the cathode material. Capacity test for the sample revealed an initial discharge capacity of 195.2 mAh·g^−1^ at 0.1 C, with capacity retention of 95.1% after 100 cycles. EIS investigation suggested that the high Li ion diffusion coefficient (3.89 × 10^−10^·cm^2^·s^−1^), caused by the mesoporous nature of the cathode powder, could be regarded as the important factor for the excellent performance of the Li_1.27_Cr_0.2_Mn_0.53_O_2_ layered material. The results demonstrated that the cathode material prepared by our approach is a good candidate for lithium-ion batteries.

## 1. Introduction

Manganese-based cathode materials have been widely studied for use in lithium-ion batteries due to their low cost, nontoxicity and, in the case of layered LiMnO_2_, high theoretical capacity (285 mAh·g^−1^). Advances in electrode materials are very important for the development of rechargeable lithium-ion batteries. As an electrode of a lithium-ion battery, several compounds such as spinel LiMn_2_O_4_, layered LiCoO_2_, and LiNiO_2_ have been extensively studied [1,2,3]. LiMnO_2_ in the form of a layered compound having the structure R-3m is interesting as a cathode material, but suffers from severe capacity decay during cycling. Besides, layered LiMnO_2_ transforms into a spinel structure during the lithium insertion/extraction process due to cation migration [4]. Recently, derivatives of layered manganese oxides, such as substituted LiM_x_Mn_1−x_O_2_ (M = Al, Co, Cr, etc.) [5,6] and lithium-saturated solid solutions or nanocomposite Li_2_MnO_3_-LiMO_2_ (M = Ni, Co, or Fe) [5,6,7,8,9,10,11], have been investigated, in order to obtain cathodes with prolonged structural integrities and enhanced electrochemical performance.

Recently, because of the high specific capacity of lithium-rich phase xLi_2_MnO_3_·(1-x)LiMO_2_ (M = Cr, Co, Mn, Ni, or Fe) [11,12,13,14,15,16,17,18] composites, it has been widely studied for the use as cathode material in lithium secondary battery. Some research results have been reported on the layered structure LiCrO_2_-Li_2_MnO_3_. It was reported that by a solution method and subsequent quenching method, Li[Cr_x_Li_(1-x)/3_Mn_2(1-x)/3_]O_2_(0.1 ≤ x ≤ 0.4) with nanocomposite structure was synthesized by Park et al. [12]. The material exhibited a high capacity of 195 mA·g^-1^, when the cutoff voltage was between 2.4V and 4.7 V and the current density was 11.98 mA·g^−1^. Kim et al. [19] synthesized Li[Cr_x_Li_(1/3-x/3)_Mn_(2/3-2x/3)_]O_2_ (0 < x < 1) by the sol-gel method. At a specific current density of 5 mA·g^−1^, Li[Cr_x_Li_(1-x)/3_Mn_2(1-x)/3_]O_2_ with x = 1/6 can exhibit a high reversible capacity of 230 mAh·g^−1^ when the voltage is between 2.0 V and 4.8 V. In addition, layered Li-Cr-Mn-O cathode materials related to the LiCrO_2_-LiMnO_2_-Li_2_MnO_3_ solid solution have been synthesized by the mixed hydroxide method [8,20]. The cathode material in terms of high capacity and stable cycling performance exhibits an average discharge capacity of 204 mAh·g^−1^ between 2.5 and 4.5 V versus Li/Li^+^. It is reported by Wu et al. [20] that the LiMnO_2_ cathode material can reduce the topographical change from orthogonal to monoclinic geometry while improving cycle performance and reversible capacity through a small amount of Cr doping. On the other hand, Ko et al. [21] reported the 0.55Li_2_TiO_3_-0.45LiCrO_2_ composite have the highest initial discharge capacity of 203 mAh·g^−1^, showing that the chromium ions can participate in the electrochemical reactions. However, almost all the materials mentioned above showed fast capacity fading during the cycling and new systems and synthesis method were required to prepare the material with high capacity and stable cycling performance.

In this work, layered Li_1.27_Cr_0.2_Mn_0.53_O_2_ powders were synthesized by a nanomilling-assisted solid-phase method. The obtainedLi_1.27_Cr_0.2_Mn_0.53_O_2_ cathode delivered high capacity close to the theoretical value with good capacity retention of 95.1% after 100 cycles. Such good electrochemical behavior can be attributed to the mesoporous nature of the cathode particles which offer fast pathway for the migration of the Li ions. The layered Li_1.27_Cr_0.2_Mn_0.53_O_2_material presented high potential as a candidate to meet the demands of LIBs with high energy density.

## 2. Experimental

### 2.1. Synthesis of Li_1.27_Cr_0.2_Mn_0.53_O_2_ Using the Solid-State Reaction Method

The Li_1.27_Cr_0.2_Mn_0.53_O_2_ cathode materials were prepared by solid reaction method. In a typical process, according to the chemical composition with certain molar ratio of Li:Cr:Mn = 1.32:0.2:0.53, lithium hydroxide, in a purity of 99%, was dissolved in distilled water. Then, Cr_2_O_3_ (99%, Aladdin, Shanghai, China) and MnO_2_ (Aladdin, 99% pure) powders were added to the above solution to form a suspension with continuous stirring. The resulting suspension was poured into a ball mill and treated for 2 h to achieve an uniform slurry with particle size of 200–300 nm using Zirconia grits with a diameter of 0.4 mm as the grinding media at a speed of 2000 rpm. Next, the ball-milled slurry was spray dried and the precursor powders were obtained. Finally, the powders were calcined at 500 °C for 3 h and then fired at 950 °C for 12 h in nitrogen to form the target Li_1.27_Cr_0.2_Mn_0.53_O_2_ cathode materials.

### 2.2. Instrumentation

The phase purity and crystal structure determination of the prepared powders were identified by X-ray diffraction (XRD, TD3500, Tongda, Dandong, China) method with Cu Kα radiation (λ = 1.54056 Ǻ) carried out at 40 kV and 30 mA. The data were collected in the 2θ range of 10 to 70°. Rietveld refinements of XRD patterns were carried out by using Jade 9 software (materials data Inc., Livermore, CA, USA). The morphologies of the samples were examined by scanning electron microscopy (SEM, Hitachi, SU8200, Tokyo, Japan). XPS measurements were carried out using an ESCALAB 250Xi spectroscopy (Thermo Fisher Scientific, Waltham, MA, American) with Al Kα radiation. The binding energy was calibrated with respect to the conductive C 1s (285.0 eV).

The electrochemical performances of the samples were investigated using electrodes CR2016 coin-type cells at 25 °C, which were assembled inside a glove box filled with Ar. In a typical process, a mixture of the calcined powders, carbon black and binder, in the appropriate weight ratio of (8:1:1) dispersed by N-methyl-2-pyrrolidone (NMP). The slurry was then cast on aluminum foil to form a sheet that was cut into a circle of 1.44 cm^2^, and then dried in vacuum at 110 °C for 10 h. The loading of Active material was 4–5 mg per disk. Lithium metal foils were used as the working anode. The Celgard 3501 membrane (Kejing, Shengzhen, China) was employed as the separator. A special electrolyte provided by Dongguan Hangsheng (Dongguan, China), which can work at high voltage, was employed in the experiment. The electrolyte is composed of 1 M LiPF6/fluoroethylene carbonate (FEC)−ethyl methyl carbonate (EMC) (3:7 in volume ratio), with 0.5 wt% LiDFOB additive [22]. The electrochemical performance of the prepared cathode materials was determined on a Land CT2001 battery tester (Lanhe, Wuhan, China) at the voltage of 2.0–4.9 V. The cyclic voltammetry (CV) studies were conducted at a scan rate of 0.0 5mV·s^−1^ with cut-off voltages of 2.0 V and 4.9 V. Electrochemical Impedance Spectroscopy (EIS) was conducted by an electrochemical workstation (Autolab Pgstat302n, Metrohm, Zofingen, Switzerland) and the data were collected in the frequency range of 0.05 to 500 KHz.

## 3. Result and Discussion

### 3.1. Phase Purity of Synthesized Cathode Powders

Figure 1a showed the XRD patterns of the cathode Li_1.27_Cr_0.2_Mn_0.53_O_2_ materials prepared at 900 °C, 950 °C, 1000 °C. The peaks can be indexed into a hexagonal α-NaFeO_2_structure (space group R-3m), except several superlattice ordering peaks between 20° and 30° marked in the figure with arrows. These superlattices are regarded as an indicator of the coexistence of Li_2_MnO_3_ phase. In an earlier report by Kim et al. [23], the existence of Li_2_MnO_3_ phase will cause the formation of MnO_3_ phase during the electrochemical process, which will release O_2_ gas during a subsequent chemical process. The evolution of O_2_ gas will reduce the stability of the cathode itself and lead to a fast decay in electrochemical performance. Therefore, the observation of superlattice in the XRD pattern can be regarded as an indicator to determine whether the sample is well prepared or not. By comparison, it is clear that the sample prepared at 950 °C is of high purity and therefore the synthesis temperature was fixed at 950 °C for the sample preparation thereafter. Furthermore, the splitting of the (006)/(012) and (018)/(110) can be observed from the pattern, indicating that the prepared cathode material has a highly ordered layered structure. The above XRD results strongly indicated the successful synthesis of a pure phase for the layered hexagonal product in the experiment, and the obtained Li_1.27_Cr_0.2_Mn_0.53_O_2_ compound can be regarded as a normal Li[Li_0.27_Cr_0.2_Mn_0.53_]O_2_ layered complex.

Rietveld refinement for the sample S950 is presented in Figure 1b. The lattice parameters of the a axis and c axis were calculated using Jade 9. The lattice parameters and atomic occupancies of the sample was provided on the basis of high symmetry R-3m space group, as listed in Table 1 and Table 2. Generally, the error indicators, R_wp_ and R in Table 1, are two important factors for evaluating the refinement results, where Rwp is the weight distribution factor and R is the confidence factor, and it is reliable and acceptable when the R factor are below 10%. The observed and calculated patterns match well, so the refinement is acceptable. Furthermore, higher values of *I_003_/I_104_* listed in Table 2 correspond to the ideal layered structure of the cathode material. It is reported by Li et al. that when the *I_003_/I_104_* exceeded 1.311, no cationic disordering was presented [24]. In our approach, *I_003_/I_104_* value of 1.625 was obtained for sample S950, suggesting the absence of cationic disordering. Such a conclusion can also be obtained from the fact that Li1 (3b) sites were not filled with Cr/Mn cations in Table 1. In addition, more pronounced splitting of the (018) and (110) doublet greatly suggested that the cathode material is composed of a well-defined layered structure, and thus good electrochemical performance for the sample material was expected.

### 3.2. Valence States of Mn and Cr Ions within the Cathode Powder

XPS investigations were conducted to confirm the valence states of the cathode powders. Since MnO_2_ and Cr_2_O_3_ were employed as raw materials, Cr^3+^ and Mn^4+^ were expected to be present in the synthesized compound. The whole pattern survey for the Li_1.27_Cr_0.2_Mn_0.53_O_2_ powder is presented in Figure 2a, where peaks representing Cr, Mn, O, and C ions were observed and denoted on the figure. Detailed investigations of the Mn2p and Cr2p signals were shown in Figure 2b,c, where the spectra were deconvolved. For the spectrum of Mn2p in Figure 2b, both peaks can only be deconvolved into one single peak, suggesting the existence of Mn^4+^ ions. For the spectrum of Cr2p in Figure 2c, similar results were observed. Both the peaks forCr2p can only be only deconvolved into one single peak, indicating the presence of Cr^3+^ ions within the synthesized cathode powder. The fitting results were listed in Table 3. It can be found from Table 3 that the valence states of Cr and Mn ions in Li_1.27_Cr_0.2_Mn_0.53_O_2_ powder are +3 and +4, respectively. XPS results show that the valence of Mn and Cr ions are consistent with the valence state of the starting raw materials, as we expected.

### 3.3. SEM Observation of the Prepared Powder

The morphologies of the prepared samples were monitored by SEM. In general, the morphologies of the spray dried sample powders are characterized by uniform and large spherical agglomerates. Figure 3 shows the SEM images of Li_1.27_Cr_0.2_Mn_0.53_O_2_ prepared at 950 °C under N_2_ atmosphere. It can be seen from Figure 3a that the powder is composed of uniform spherical particles with diameter of 2–4 μm. Figure 3b shows the enlarged micrograph of an individual particle, where tiny primary grains with size approximately 100–500 nm were observed. Microspores among the nanosized grains were also observed. N_2_ adsorption/desorption test was conducted to get more details about the mesostructure. At the liquid nitrogen temperature, in the nitrogen-containing atmosphere, the surface of the powder will physically adsorb nitrogen. The specific surface area (S_BET_) of the powder can be obtained by the following formula.
$$S_{BET}=4.36V/W$$
where *V* is the adsorption amount of a complete layer of nitrogen molecules adsorbed on the surface of the powder. *W* is weight of sample. The adsorption/desorption plot and the size distribution of the nanopore within the mesostructure are presented in Figure 3c,d. The specific surface area (S_BET_) can be calculated and the value was determined as 25.88 m^2^·g^−1^ for S950. An average diameter of 4–5 nm for the individual nanopore was also confirmed from Figure 3d. Due to the presence of this stable three-dimensional framework and mesoporous properties, the S950 sample will exhibit excellent cycle stability during charge–discharge. The numerous nanoparticles which aggregate to form porous microspheres can also enhance the transmission of Li ions by providing a short path for the intercalation/deintercalation of lithium-ions. It is possible to observe mesopores interconnected in the primary particles, which is advantageous for increasing the transport of Li^+^ in the microspheres through the liquid electrolyte. Such nano-micro structures of the spherical cathode powders provide both short migration pathway and improved surface area for the redox process and thus good electrochemical performance could be expected.

### 3.4. Electrochemical Performance

The electrochemical performances of the samples prepared at different temperatures were compared. The charge–discharge curves at the first cycle are presented and compared in Figure 4. It can be seen from the charging curves that, all three charging curves showed a gradually declined tendency, no obvious charging plateau was observed. Such charging curves greatly suggested the solid solution transition during the electrochemical process, which is very similar to the behavior of the other families of LiMO_2_ layered structures [25]. Furthermore, it is also clear from the figure that the powder fired at 950 °C offered a highest specific capacity of 195.2 mAh·g^−1^. According to the formula of Li_1.27_Cr_0.2_Mn_0.53_O_2_, if 0.2 mole Cr^3+^/Cr^6+^ redox pairs were employed in the electrochemical process, 0.6 mole electrons will be released, and the nominal cathode compound will present a theoretical capacity of 200.18 mAh·g^−1^. Therefore, in our experiment, by using Cr_2_O_3_ andMnO_2_ as the starting materials, electrochemical cycles using Cr^3+^/Cr^6+^ redox pairs, rather than the Mn redox pairs, were achieved with specific capacity close to its theoretical value.

To confirm the solid solution transition manner of the electrochemical process, XRD investigations of the electrodes at charging and discharging states were compared to the pristine electrode before the capacity test, as shown in Figure 5. It is clear from Figure 5a that similar patterns were obtained for electrode samples at different state of charging. The pattern for the pristine electrode before cycling (pattern a1) showed good agreement to the XRD pattern of the Li_1.27_Cr_0.2_Mn_0.53_O_2_powder sample, indicating the hexagonal nature of the cathode material before electrochemical cycling. Pattern a2 and pattern a3 showed the XRD results after the first charging/discharging process. All three patterns looked very alike, implying the same crystal structure of hexagonal type throughout the electrochemical process. Figure 5b showed an enlarged pattern at 2θ range of 43° to 46°. It is clear from the figure that no diffraction peak splitting was observed, only slight peak position shift was presented. Such results confirmed the solid solution transition manner of the cathode material during the Li intercalation and deintercalation process, rather than the two-phase transition manner, which is usually judged by the peak splitting.

XPS investigations of the electrodes at different SOC were conducted to further confirm the valence of chromium ions and manganese ions during the cycling. Figure 6 shows the changes in the valence state of Cr and Mn ions during charge and discharge processes. Detailed investigations of the Mn2p signals were shown in Figure 6a and Figure 6c, where the spectra were deconvolved. Figure 6a,c shows two main peaks in the Mn2p spectra, which can be assigned to the manganese 2p_3/2_ at 641.8 eV and 2p_1/2_ at 654.0 eV. The binding energy is well consistent with the data on MnO_2_, meaning Mn^4+^ in the Li_1.27_Cr_0.2_Mn_0.53_O_2_. Therefore, as can be seen from Figure 6a,c, the Mn ions remain at +4 valance states and do not participate in the electrochemical reactions during charging and discharging process. However, for Cr2p signals, the peaks can be deconvolved into two peaks, indicating the presence of Cr^3+^ ions after charging and Cr^6+^ ions after discharging, as can be seen from the curves in Figure 6b,d. Therefore, by comparison, it can be deduced that Cr^3+^/Cr^6+^ redox pairs played the dominant role in the electrochemical process.

The fitting results from software Avantage were listed in Table 4. It is quite clear from Table 4 the Mn ions were kept very stable as +4 in valance state for both samples regardless charging or discharging state and no other valance state for Mn ions were detected. However, the difference in the valence state of Cr ions was obvious. For the sample after charging, most Cr ions were kept as +6, while it were kept as +3 for sample after discharging. Only a small amount of Cr^3+^ ions (~4 atm%) were observed in the charged electrode and a small portion ofCr^6+^ ions (~2.3 atm%) were detected in the discharged electrode. Therefore, from the above result, it is clear that Cr^3+^/Cr^6+^redox pairs, rather than theMn^4+^/Mn^3+^redox pairs, decided the electrochemical performance of the prepared cathode material.

The voltammetric behavior of the Li_1.27_Cr_0.2_Mn_0.53_O_2_ electrodes after 1, 2, and 50 cycles in the voltage range of 2.5 to 5.0 V are presented in Figure 7. It is clear that all the three curves are very similar, and the noticeable anodic peak around 3.6 V and cathodic peak around 3.2 V for the first cycle were observed. For the Cr^3+^/Cr^6+^ redox couple in other system, such as NaCrO_2_, the anodic scan showed a sharp and intense peak at 3.03 V and a low intensity peak at 3.3 V [26]. The corresponding cathodic peaks are observed at 2.8 V and 3.25 V, respectively. These peaks were associated with Cr^3+^/Cr^4+^ redox couple besides the Cr^3+^/Cr^6+^ redox couple. In some other compounds, the cyclic voltammograms of LiCr_0.05_Ni_0.45_Mn_0.5_O_2_ and LiCr_0.1_Ni_0.4_Mn_0.5_O_2_ [10] revealed oxidation peaks in 2.9, 3.9, and 4.4 V regions and reduction peaks in the 2.7 and 3.6 V regions; additionally, more complicated phase transformation was observed. The same happened in our case, an oxidation peak at 4.5 V and a reduction peak at 4.13 V were also observed in the CV profile, which indicates that Cr ions in other valance state could be involved in the electrochemical process. The coexistence of multivalance state of Cr ions implies the complex of the electrochemical process, in which the transitions between Cr^3+^/Cr^4+^/Cr^6+^ could take place during charging and discharging processes.

Furthermore, it can be found form the figure that at the second run, the anodic peak shifts to the negative potential of 3.4 V, while the cathodic peak shifted to the potential of 3.3 V, a very small ΔV(0.1V) was observed for the prepared Li_1.27_Cr_0.2_Mn_0.53_O_2_ electrode. For the runs thereafter until the 50th run, the redox pair at 3.3/3.4 V kept unchanged. This reinforces the conclusion that the material does not transform into other structures (like spinel in other layered phase) during cycling [18,27,28]. Such electrochemical behavior indicates that stable cycling has been set up within the electrode and therefore high performance of the electrodes can be expected.

Figure 8a shows the charge–discharge profiles with the cutoff voltages of 2.0 and 4.9 V at a rate of 0.1 C. The initial discharge specific capacity of Li_1.27_Cr_0.2_Mn_0.53_O_2_ was found to be 195.2 mAh·g^−1^ at the first cycle. It increased gradually onto 205.4 mAh·g^−1^ at the 10th run and then slightly dropped down at subsequent runs. With further cycling, both the capacity and the voltage profile became stable, which indicated that the structure is stable over the entire intercalation–deintercalation process. In addition, the degree of polarization gradually decreases with the increased cycles and it became relatively stable until the 10th cycle. The above results greatly implied that the Li_1.27_Cr_0.2_Mn_0.53_O_2_ cathode material presents excellent cycling stability at room temperature, regardless the difference in the voltage profiles. On the other hand, no plateau was observed at ~2.9 V on the profile, which indicates that the prepared material does not convert into a spinel phase, which was reported by other groups [18]. Figure 8b shows the cyclic performance of the Li_1.27_Cr_0.2_Mn_0.53_O_2_ cathode material. The discharging capacity increased up to 205.4 mAh·g^−1^ in the 10th cycle and then was maintained almost constant beyond the subsequent cycles. Moreover, it can be seen from Figure 8b that the capacity retention was 95.1% during the 100th cycles. All the results suggested an outstanding electrochemical performance of the prepared cathode.

The electrochemical impedance spectra of Li_1.27_Cr_0.2_Mn_0.53_O_2_ measured after 1, 30, and 100 cycles are shown in Figure 9 in an attempt to figure out the reason for the high performance of the material. The resistivity of the sample is represented by the intercept of the semicircle and the real axis (Z’). In the Nyquist representation, due to the interfacial migration of lithium-ions between the surface layer and the electrolyte, the impedance data will appear in a semicircular form in the high-frequency region. [29]. The semicircle reduced significantly from 1st to 100th cycle of discharge. Wu et al. indicates that the reduction of charge transfer resistance can effectively accelerate the kinetics of the intercalation–deintercalation process, which is shown by a significant reduction in the semicircle in the figure [18]. Such impedance data can be simulated with an equivalent circuit and the solution resistance and the charge transfer resistance could be obtained. A suitable equivalent circuit containing Rs, Rct, CPE, and Zw showed the best fitting results for the experimental data, and the output results for Rs and Rct are listed in Table 5. For the value of Rs, a recording of only several ohms was obtained with a very slight change, since it represents for the resistance of the electrolyte itself. However, in term of Rct, the value varies from 233.88 Ω to 91.86 Ω, which indicates an obvious decrease in interfacial resistance. Hence, in our case, high specific capacity with stable columbic efficiency for the sample S950 was obtained during the cycling.

In addition, the following equation expresses the calculation equation of the diffusion coefficient of lithium-ions (D_Li_^+^) [30,31].
(1)$$D_{Li^{+}}=\frac{0.5R^{2}T^{2}}{A^{2}F^{4}C^{2}\sigma_{W}^{2}}$$
where *A* the surface area of the electrode, *T* the absolute temperature, *F* the Faraday constant, *R* is the gas constant, *C* the concentration of Li^+^ in the material, and *σ_w_* is the Warburg factor obeying the following relationship,
(2)$$Z_{Re}=R_{S}+R_{ct}+\sigma \omega^{-0.5}$$
*σ_w_* can be obtained by fitting of the real parts of impedance Z_Re_ vs. ω^−0.5^ in the frequency range of 0.05 Hz to 1 Hz as shown in the Figure 10. As a result, the diffusion coefficients of the Li_1.27_Cr_0.2_Mn_0.53_O_2_ material calcined at 900 °C (1.05 × 10^−10^ cm^2^·s^−1^), 950 °C (3.89 × 10^−10^ cm^2^·s^−1^), and 1000°C (1.34 × 10^−10^ cm^2^·s^−1^) were determined. Furthermore, by calculation, the diffusion coefficient of the material S950 after 100 cycles (3.89 × 10^−10^ cm^2^·s^−1^) is higher than the values of Li_1.2_Ni_0.2_Mn_0.2_O_2_ (1.63 × 10^−12^ cm^2^·s^−1^) and Li_1.23_Mn_0.46_Ni_0.15_Co_0.16_O_2_ (2.78 × 10^-15^ cm^2^·s^−1^) [31] reported by others. Such a high diffusion coefficient are supposed to provide a fast Li diffusion within the layered Li_1.27_Cr_0.2_Mn_0.53_O_2_ based on the Cr^3+^/Cr^6+^ redox repairs, and high potential of the material as a cathode for LIBs could be achieved.

From the above results, it can be found that the sample S950 presented the best electrochemical performance among the three samples. The S950 cathode delivered an initial discharge capacity of 195.2 mAh·g^−1^ with capacity retention of 95.1% after 100 cycles. All the results can be explained by the fact that a fast Li ion transportation was achieved since the largest Li ion diffusion coefficient was observed for the sample.

## 4. Conclusions

Li_1.27_Cr_0.2_Mn_0.53_O_2_ cathode materials with hexagonal α-NaFeO_2_ structure (space group R-3m) were successfully prepared by a nanomilling-assisted solid-state reaction method. The initial XPS studies confirmed the existence of Cr^3+^ and Mn^4+^ in the prepared powders as expected. The Li_1.27_Cr_0.2_Mn_0.53_O_2_ electrode showed a typical solid solution transition during the electrochemical cycling and no other side process was observed. CV tests revealed an oxidation/reduction couple of 3.6/3.2 V ascribed to the Cr^3+^/Cr^6+^ redox pair rather than the Mn redox pairs. XPS studies on the electrodes after charging/discharging further confirmed that the triple-electron process of the Cr^3+^/Cr^6+^ redox pair, not the Mn redox pairs, was superior to the electrochemical process within the cathode material. The capacity test for the sample exposed a high reversible capacity of ~195.2 mAh·g^−1^ (close to its theoretical value) between 2.0 and 4.9 V at 0.1 C, with capacity retention of 95.1% after 100 cycles. EIS investigation revealed a high diffusion coefficient of 3.89 × 10^−10^ cm^2^·s^−1^, which can be regarded as the reason for the good electrochemical performance of the prepared cathode material. As a whole, the prepared compound presented high potential as a candidate for the use as cathode in LIBs.

## Figures and Tables

**Figure 1 materials-12-00468-f001:**
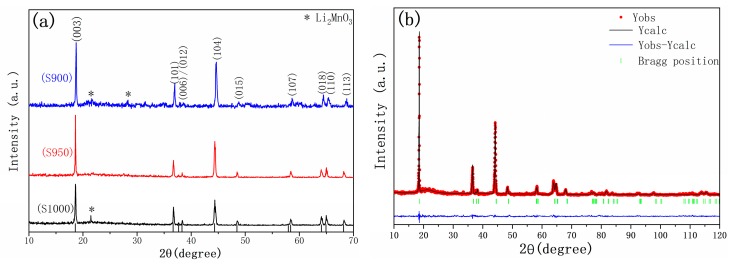
(**a**) X-ray diffractions of samples prepared. (**b**) Rietveld refinement results for sample S950.

**Figure 2 materials-12-00468-f002:**
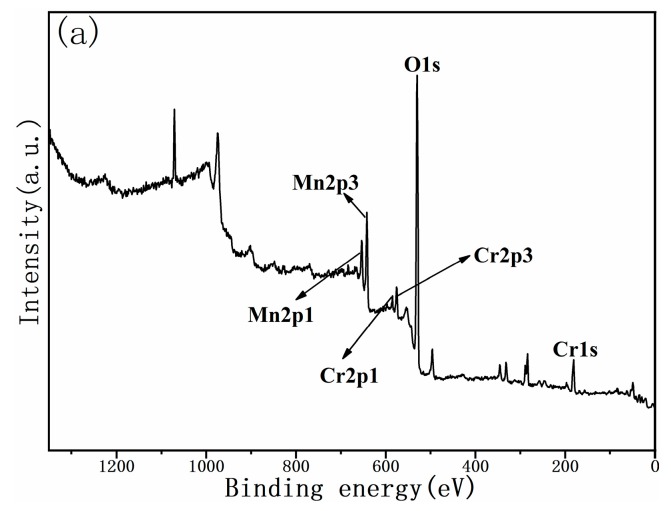
XPS spectra of the S950 powder (**a**). Whole pattern survey: (**b**) Mn2p and (**c**) Cr2p.

**Figure 3 materials-12-00468-f003:**
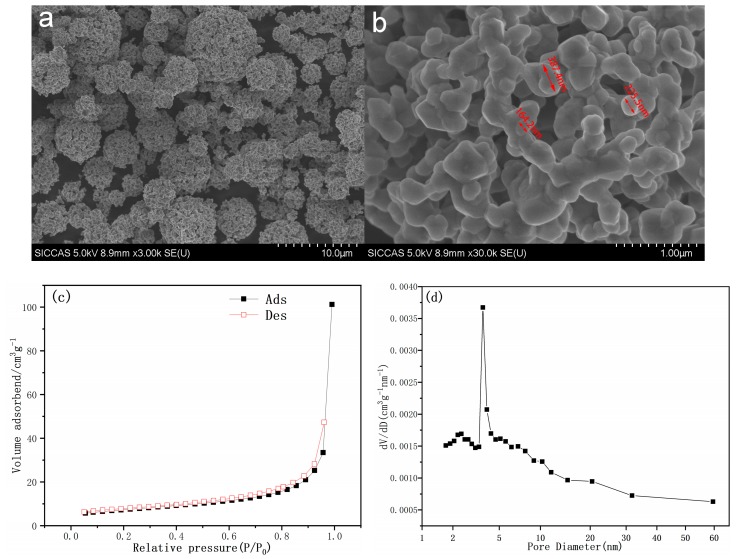
(**a**,**b**) SEM powder overview and individual particle, (**c**) nitrogen adsorption/desorption isotherms, and (**d**) mesopore distribution of Li_1.27_Cr_0.2_Mn_0.53_O_2._

**Figure 4 materials-12-00468-f004:**
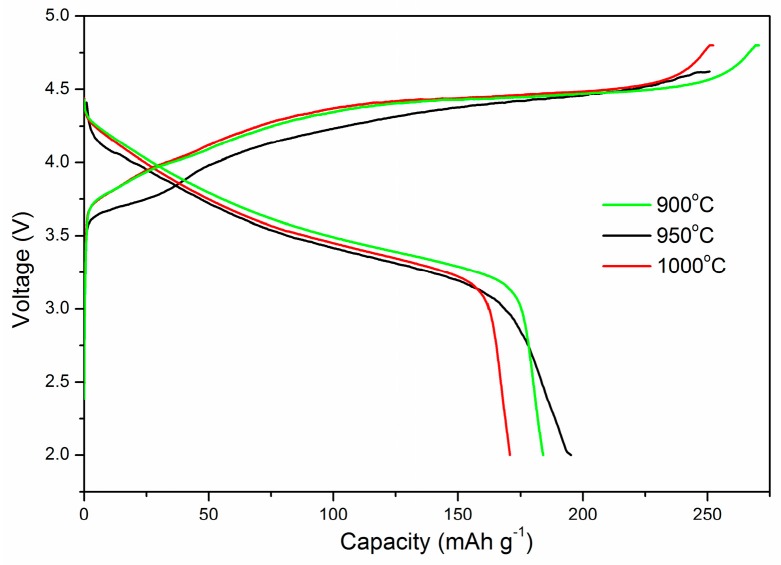
The initial charge–discharge curves of samples sintered with different temperature.

**Figure 5 materials-12-00468-f005:**
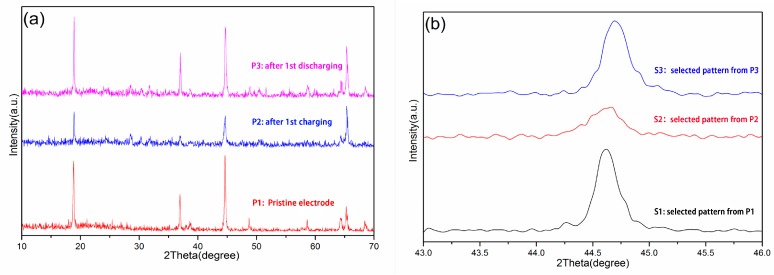
XRD for the electrodes: (**a**) whole range pattern and (**b**) at selected range.

**Figure 6 materials-12-00468-f006:**
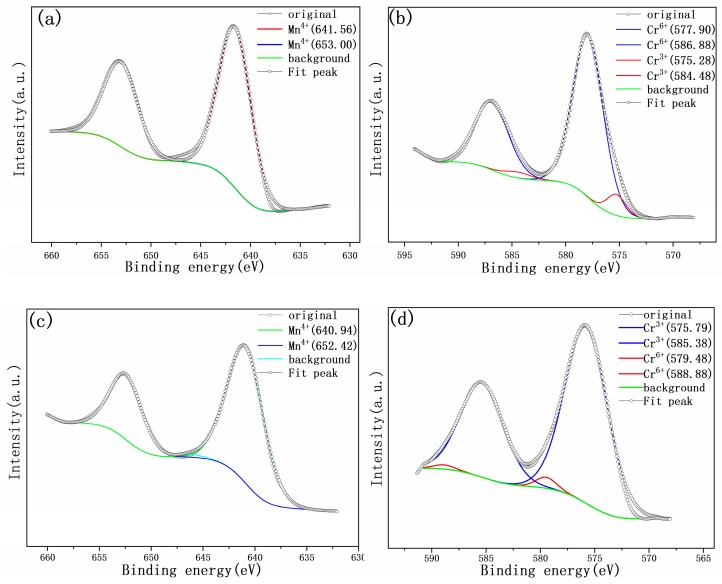
Deconvolved XPS spectra of electrodes at different SOC. (**a**) Mn2p at charging state, (**b**) Cr2p at charging state, (**c**) Mn2p at discharging state, and (**d**) Cr2p at discharging state.

**Figure 7 materials-12-00468-f007:**
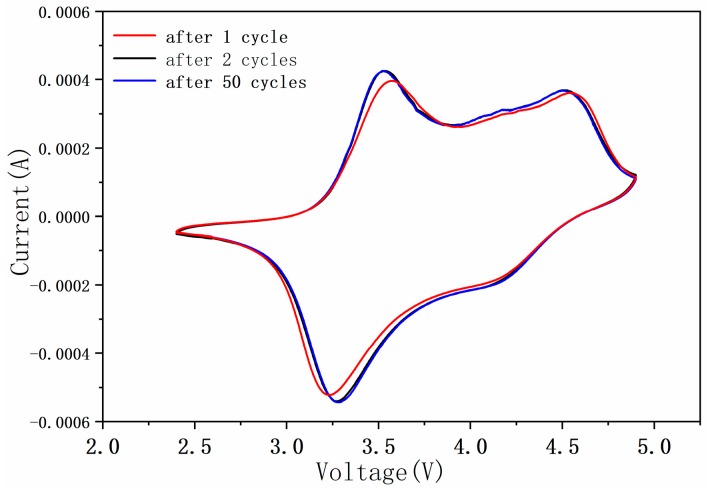
Cyclic voltammetry (CV) for the Li_1.27_Cr_0.2_Mn_0.53_O_2_ cell obtained at a scan rate of 0.05 mV·s^−1^.

**Figure 8 materials-12-00468-f008:**
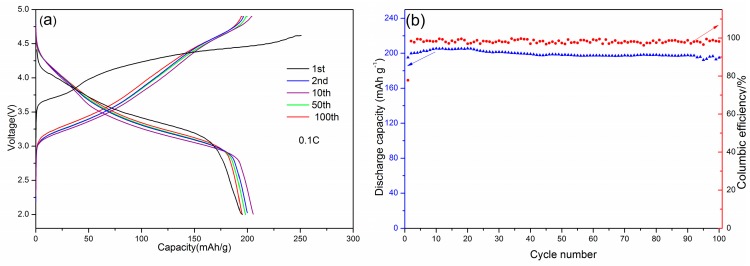
Charge–discharge curves and columbic efficiency of Li_1.27_Cr_0.2_Mn_0.53_O_2_ (**a**,**b**).

**Figure 9 materials-12-00468-f009:**
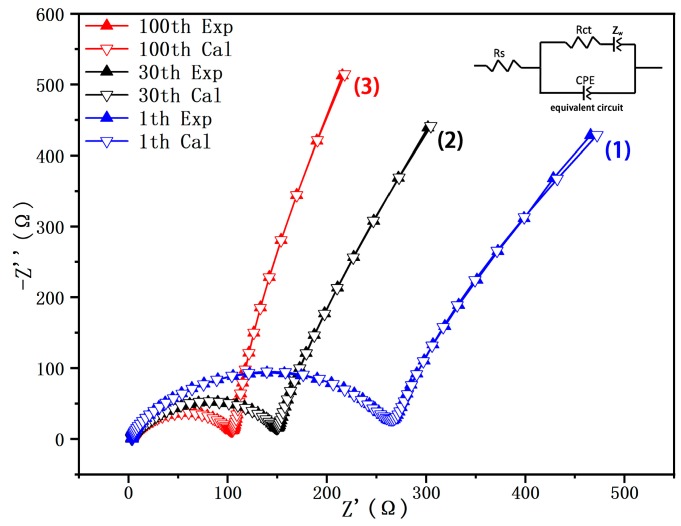
The impedance spectra for the Li_1.27_Cr_0.2_Mn_0.53_O_2_ electrode: (1) after first cycle, (2) after 30th cycle, and (3) after 100th cycle.

**Figure 10 materials-12-00468-f010:**
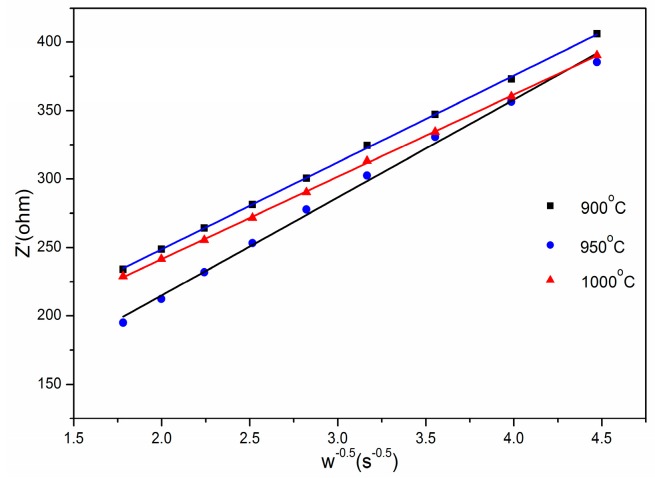
Plots comparison of Z’ vs. ω^−0.5^ for Li_1.27_Cr_0.2_Mn_0.53_O_2_ after 100 cycles.

**Table 1 materials-12-00468-t001:** Rietveld refinement results for the XRD patterns.

Sample	Atom	Wyckoff Position	x	y	z	Occupancy	R (%)	Rwp (%)
S950	Li (1)	3b	0	0	0.5	0.9998	9.49	10.44
	Li (2)	3a	0	0	0	0.2686		
	Mn (1)	3a	0	0	0	0.5274		
	Cr (1)	3a	0	0	0	0.1984		
	O (1)	6c	0	0	0.2416	1		

**Table 2 materials-12-00468-t002:** Rietveld refinement results of lattice parameters for S950.

Sample	a(Å)	c(Å)	V(Å^3^)	*I_003_*/*I_104_*
S950	2.8643	14.2645	101.35	1.625

**Table 3 materials-12-00468-t003:** XPS simulation results for Li_1.27_Cr_0.2_Mn_0.53_O_2_ powder sintered at 950 °C.

Source	Component	Valance State	BE/eV	FWHM/eV	Relative Area/%
Figure 2b	642.5	Mn^4+^	642.1	1.9	100
		Mn^3+^	-	-	0
Figure 2b	653.4	Mn^4+^	653.37	2.1	100
		Mn^3+^	-	-	0
Figure 2c	576.5	Cr^3+^	576.49	1.96	100
		Cr^6+^	-	-	0
Figure 2c	586.8	Cr^3+^	586.72	2	100
		Cr^6+^	-	-	0

**Table 4 materials-12-00468-t004:** X-ray photoelectron spectroscopy (XPS) simulation results for the electrode.

Source	Component/eV	Valance State	BE/eV	FWHM/eV	Relative Area/%
Figure 6a Charged electrode	641.55	Mn^4+^	641.56	1.71	100
		Mn^3+^	-	-	0
Figure 6a Charged electrode	653.00	Mn^4+^	653.03	1.55	100
		Mn^3+^	-	-	0
Figure 6b Charged electrode	578.01	Cr^6+^	577.85	1.73	95.8
		Cr^3+^	575.19	1.08	4.2
Figure 6b Charged electrode	587.43	Cr^6+^	586.97	1.93	96.2
		Cr^3+^	584.68	1.49	3.8
Figure 6c Discharged electrode	641.01	Mn^4+^	640.94	1.34	100
		Mn^3+^	-	-	0
Figure 6c Discharged electrode	652.42	Mn^4+^	652.45	1.76	100
		Mn^3+^	-	-	0
Figure 6d Discharged electrode	576.21	Cr^6+^	579.48	0.58	2.5
		Cr^3+^	575.79	1.75	97.5
Figure 6d Discharged electrode	585.7	Cr^6+^	588.88	0.53	2.1
		Cr^3+^	585.38	1.79	97.9

**Table 5 materials-12-00468-t005:** Electrochemical impedance spectroscopy (EIS): initial resistance (Rs) and charge transfer resistance (Rct) after cycling.

Cycles	Rs(Ω)	Rct(Ω)
1th cycle	3.50	233.88
30th cycle	2.64	132.88
100th cycle	3.34	91.86

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
