# Peer review of "Enhanced Electrochemical Performance of Li1.27Cr0.2Mn0.53O2 Layered Cathode Materials via a Nanomilling-Assisted Solid-state Process"

_materials, 2019, doi:10.3390/ma12030468_

Round 1
Reviewer 1 Report
The authors selected 3 different temperatures (900, 950 and 1000 degrees) to anneal the samples and compared their electrochemical performance. I do not think the results are convictive because the gaps between the temperatures are too small.
Author Response
Thanks for your comment.
The temperature we choose for the calcination of the sample powders was suggested from the references listed below:
Ref1: Park, C. W.; Kim, S. H.; Nahm, K. S.; Chung, H. T.; Lee, Y. S.; Lee, J. H.; Boo, S.; Kim, J., Structural and electrochemical study of Li[CrxLi(1−x)/3Mn2(1−x)/3]O2 (0≤x≤0.328) cathode materials. J. Alloys Compod. 2008, 449, (1-2), 343-348
Ref2: Wu, X.; Chang, S. H.; Park, Y. J.; Ryu, K. S., Studies on capacity increase of Li1.27Cr0.2Mn0.53O2-based lithium batteries. J. Power Sources 2004, 137, (1), 105-110.
Ref3: Kim, K. S.; Lee, S. W.; Moon, H. S.; Kim, H. J.; Cho, B. W.; Cho, W. I.; Choi, J. B.; Park, J. W., Electrochemical properties of Li–Cr–Mn–O cathode materials for lithium secondary batteries. J. Power Sources 2004, 129, (2), 319-323.
Ref4: Ko, Y. N.; Kim, J. H.; Lee, J. K.; Kang, Y. C.; Lee, J. H., Electrochemical properties of nanosized LiCrO2·Li2MnO3 composite powders prepared by a new concept spray pyrolysis. Electrochim. Acta 2012, 69, 345-350
In reference1, the authors provided a quenching method to prepare the target cathode Li[CrxLi(1−x)/3Mn2(1−x)/3]O2 material which was annealed under temperature 900oC.
In reference2, the authors reported a solid state process for the preparation of Li1.27Cr0.2Mn0.53O2-cathode material, in which the precursors were calcinated under temperature of 900oC.
In reference3, the Li–Cr–Mn–O cathode materials were obtianed by calcinating the precursor at temperature from 650-950oC.
In refrence4, the LiCrO2·Li2MnO3 composite cathode powders were prepared by a spray pyrolysis process which were operated under 1000 oC.
All th erefences have been cited in the original submssions. We think that, for a solid state reaction, high calcinating temperature helps the ionic diffusion and promotes the phase transition. Low temperatures below 900 oC were not taken into account in the experiment since an inclusive diffraction peak was detected in figure1 with sample fired under 900 oC. Therefore, according to the references above, we set the firing temperature from 900oC to 1000oC. Temperatures over 1000oC were not considered because some other diffraction peaks were detected on the XRD pattern for sample fired under 1000 oC.
So by comparison between the XRD patterns, a suitable temperature of 950 oC was confirmed in our case for the preparation of the target cathode sample.

Reviewer 2 Report
What electrolyte did the authors use?
The applied voltage is very high (4.9 V). Most organic electrolytes break down rapidly when such high potential is applied. How did the authors mitigate this?
If the authors discuss porosity, they cannot use microscopy imaging to assess pore dimensions. They need techniques such as gas sorption to quantify surface areas and pore dimensions, as SEM cannot describe the nanopores and micropores that the authors mention (0.5 - 10 nm length scales)
Author Response
Thanks for your efforts made on the manuscript.
As for the eletrolyte, a specially designed electrolyte from the supplier, Donguan Hangsheng company, rather than a commercial product, was used in the experiment. The electrolyte experienced 100 cycles from 2.0V to 4.9V in the experiment with a high capacity retention of 95.1%, which showed great potential in future application. Some organic additives offer the electrolyte high voltage resistance. Due to the commercial rules, we are sorry that no more details about the electrolyte be released without the permission of the company.
Related revisions on the type of electrolyte were made on page3 in red in the new submission.
For the porosity measurement, major revisions were made with the N2 abosorption experimental data, which were added in the new version of the manuscript as Figure4c and Figure4d. The surface area was determined as 25.88 m2g-1 and a pore size around 4-5nm was also confrimed.
Please refer to the revisions on page6 and page7 in red.
Reviewer 3 Report
The manuscripts describes Li1.27Cr0.2Mn0.53O2 prepared by nano milling assisted solid state process. Proposing Li1.27Cr0.2Mn0.53O2 as a cathode material isn't a new idea, as cited in the paper, but the preparation rout allowed authors to obtained material with enhanced properties. Even though the results presented in the paper seems interesting, the authors have to address the following remarks before the manuscript could be published in Materials:
- The synthesis procedure have to be described in more detail. The description lacks of parameters of all stages of synthesis (milling, drying, heat treatment). As well, the "Instrumentation" part should be extended with cell assembly details.
- Authors claim that all the peaks can be indexed into a hexagonal structure except several superlattice ordering peaks (94) - lines should be properly indexed. The reason of choosing S950 for further studies (96) does not have a support in presented XRD results unless the purity of samples indeed vary.
- When studying the valence number of manganese using XPS Authors should consider the analysis of the Mn 3s peaks. This should provide more convincing proof of the manganese valence state at all SOC.
- The grains presented on SEM micrographs seems bigger than particle sizes mentioned in the text. Authors should once more analyze those results. Moreover, the more visible markers on micrographs would help.
- Could authors provide more details on the porosity of this material and its surface area?
- The voltammetric results have to be analyzed in more details. It is clear from the graph that more is happening in this electrochemical process than the provided discussion describes.
- The EIS studies cannot be published in presented form. There is no information of equivalent circuit proposed for this results, fitting details, values of changes in resistivity. Those results should support claims in the text but there is no information on how they where analyzed. The Z plots graph should contain results of S950 after 100 cycles. The frequency region used in diffusion coefficient calculations is out of the one mentioned in experimental section.
- The English language should be revised. Especially 'results and disscusion' part contains unclear and confusing phrases like: "It is clear from Fig.6a that similar patterns were 175 obtained for sample with different treatment. The pattern for the pristine electrode before cycling 176 (pattern a1) seemed very similar to the XRD pattern of the powder sample, and hexagonal structure 177 was determined." Furthermore, in XPS part of charged and discharged materials the sample naming should be clearer. The text should be also revised in therms of editing. All wrong indexing, lack of spaces, double spaces etc. should be corrected.
Author Response
Dear referree:
Thanks for your comments on the manuscript.
Please find in the attached word file to check the reply to your suggestions.
with best wishes
Chengkang Chang

Round 2
Reviewer 1 Report
The authors have answered the questions put forward by the reviewers.
Author Response
Thanks for your suggestion. We checked the ms. for possible grammar mistakes/typo errors and made related revisions in red.
Reviewer 3 Report
Dear authors,
Thank you for correcting and improving the manuscript in accordance to my suggestions. I find that the revised version of the manuscript fulfill standards to be published in "Materials". Before final publication I recommend few further corrections:
- Please, explaining what "special electrolyte provided by Dongguan Hangsheng" means, or at least provide the information of major ingredients in this system, so readers could assume what part of cell parameters comes from electrode material and which from electrolyte itself.
- Moreover, please provide the experimental details of adsorption/desorption studies if you added those results to manuscript.
Best regards
Author Response
Point 1: Please, explaining what "special electrolyte provided by Dongguan Hangsheng" means, or at least provide the information of major ingredients in this system, so readers could assume what part of cell parameters comes from electrode material and which from electrolyte itself.
Response 1:
The electrolyte is composed of 1 M LiPF6/fluoroethylene carbonate(FEC) −ethyl methyl carbonate (EMC) (3:7 in volume ratio), with 0.5 wt % LiDFOB additive.
Lan Xia, Saixi Lee, Yabei Jiang, Yonggao Xia, George Z. Chen and Zhaoping Liu, Fluorinated Electrolytes for Li-Ion Batteries: The Lithium Difluoro(oxalato)borate Additive for Stabilizing the Solid Electrolyte Interphase, ACS Omega 2017, 2, 8741−8750
Please refer to the revised version on page 3 in red.
Point 2: Moreover, please provide the experimental details of adsorption/desorption studies if you added those results to manuscript.
Response 2:
Thanks for your suggestion. The experimental details of adsorption/desorption studies was added as follows:
At the liquid nitrogen temperature, in the nitrogen-containing atmosphere, the surface of the powder will physically adsorb nitrogen, the specific surface area (SBET) of the powder can be obtained by the following formula:
Where V is the adsorption amount of a complete layer of nitrogen molecules adsorbed on the surface of the powder. W is weight of sample.
Please check the revised version on page 5 in red.